# Improving germline transmission efficiency in chimeric chickens using a multi-stage injection approach

Danial Naseri[1], Kianoush Dormiani[2], Mehdi Hajian[1], Farnoosh Jafarpour[1], Mahboobeh Forouzanfar[2], Naeimeh Karimi[2], Mohammad Hossein Nasr-Esfahani[1,2]*

**1** Department of Reproductive Biotechnology, Reproductive Biomedicine Research Center, Royan Institute for Biotechnology, ACECR, Isfahan, Iran, **2** Department of Molecular Biotechnology, Cell Science Research Center, Royan Institute for Biotechnology, ACECR, Isfahan, Iran

* mh.nasr-esfahani@royaninstitute.org

**Data Availability Statement:** All relevant data are within the paper and its Supporting information files.

## Abstract

Although different strategies have been developed to generate transgenic poultry, low efficiency of germline transgene transmission has remained a challenge in poultry transgenesis. Herein, we developed an efficient germline transgenesis method using a lentiviral vector system in chickens through multiple injections of transgenes into embryos at different stages of development. The embryo chorioallantoic membrane (CAM) vasculature was successfully used as a novel route of gene transfer into germline tissues. Compared to the other routes of viral vector administration, the embryo's bloodstream at Hamburger-Hamilton (HH) stages 14–15 achieved the highest rate of germline transmission (GT), 7.7%. Single injection of viral vectors into the CAM vasculature resulted in a GT efficiency of 2.7%, which was significantly higher than the 0.4% obtained by injection into embryos at the blastoderm stage. Double injection of viral vectors into the bloodstream at HH stages 14–15 and through CAM was the most efficient method for producing germline chimeras, giving a GT rate of 13.6%. The authors suggest that the new method described in this study could be efficiently used to produce transgenic poultry in virus-mediated gene transfer systems.

## Introduction

Genetic modification of avian species provides great benefits for industrial and medical purposes by enabling resistance to zoonotic diseases, improving meat and egg productivity, and generating effective bioreactor models [1,2]. Despite these great benefits, the unique reproductive system of avian species has restricted the application of efficient transgenesis protocols routinely employed in mammals, so different strategies have been developed to produce transgenic birds [3,4]. Although great advances have been made in gene transfer methods, avian transgenic technology has not been widely used until recently mainly because of the relatively low and variable rates of germline transmission (GT) [2,5]. In addition to this limitation, many other challenges lie ahead, such as high embryonic lethality following injection of transgenes [5], the need to wait for two generations before having ubiquitously expressing

**Funding:** This work was supported by the Royan Institute of Biotechnology (https://www.royan.org/) (grant number 96000275). The funder had no role in study design, data collection and analysis, decision to publish, or preparation of the manuscript.

**Competing interests:** The authors have declared that no competing interests exist.

transgenic birds [3], insertional mutagenesis due to the lack of precise control over transgene integration site (especially in direct and virus-mediated gene transfer systems), and uncontrollable overexpression of transgenes [5,6]. Despite these above-mentioned limitations, a sciento-metric analysis was conducted and the results showed a recent increase in publications on transgenic poultry research [7], indicating the real possibility of application of this technology in various fields in future.

Numerous studies have attempted to produce genetically-modified transgenic birds by transducing blastoderm embryos at Eyal-Giladi and Kochav (EGK) stage X [8] using retroviruses. However, the transduction efficiency of $G_0$ founders has been reported to be low with a limited rate of GT [9–13]. To further improve the efficiency of this system, some research groups have used lentiviral vectors [14–21]. The GT efficiency using lentiviral vectors was low in these reports, ranging between 0.6 and 4.0%, although a relatively higher GT (average 17.8%) has also been reported by a research group following injection of lentiviral vectors into the subgerminal cavity of newly laid eggs [22].

To optimize the GT, some researchers have attempted to determine the ideal developmental stage for viral vector injection in chicken embryos. For example, Kawabe et al. [23] reported that the highest rate of transduction of germline cells could be obtained when the viral solution was injected directly into the blood stream of chicken embryo after 55 h of incubation at Hamburger-Hamilton (HH) stages 14–15 [24]. Likewise, through direct injection of a lentiviral vector into the blood vessels of quail embryos at similar stages (HH 13–15), Zhang et al. [25] achieved a GT efficiency of 13.0%, nearly 8 times higher than that achieved through the conventional method of blastoderm injection. Both studies proposed that circulating primordial germ cells (PGC) in early embryos are effective viral targets for generating transgenic birds using retroviral vectors.

PGC-mediated gene transfer has been suggested to be the most appropriate strategy to overcome the low GT efficiency in avian species [26–29]. Different studies have reported that *ex vivo* manipulated chicken PGC show much more improvement in GT efficiency than do the viral-mediated systems when introduced into the bloodstream of recipient embryos [30–35]. However, the processes of isolation, culture, and genetic modification of PGC, as well as screening of genome-edited PGC, are time-consuming and complicated procedures for routine use which require highly skilled operators [36].

In this study, we present the first successful use of chick embryo chorioallantoic membrane (CAM) vasculature as an accessible route for germline gene transfer. We also assessed the germline targeting efficiency of a multi-stage injection through three different routes of lentiviral vector administration. Since the multi-stage injection approach requires the eggshells to be opened at different times during the development process without compromising embryo survival, here we established a novel procedure for eggshell windowing at the desired time points during embryonic development.

## Materials and methods

### Ethics statement

All the procedures for the handling and treatment of chickens in this study were approved by the Royan Institute Animal Ethics Committee (No. R-084-2003). All experimental chickens were housed under optimum conditions of light and temperature with regular cleaning. In addition, chickens were provided ad libitum access to fresh food and water.

## Virus particles production and titration

To make HIV-based lentiviral vector particles encoding enhanced green fluorescent protein (EGFP), confluent human embryonic kidney (HEK293T) cells were co-transfected with pWPXL (Addgene, Cambridge, MA, USA), and a mixture of packaging helper plasmids (psPAX, pMD2G, Addgene). Transfection was achieved using Lipofectamine 2000 (Invitrogen, Carlsbad, CA, USA) according to the protocol provided by the manufacturer. The supernatants containing viral particles were harvested at 48 and 60 h after transfection and filtered through 0.45 μm cellulose acetate filters (Millipore, USA). Afterwards, the lentiviral particles were concentrated by ultracentrifugation at 21000 rpm for 2 h at 4°C. The resulting viral pellet was resuspended in an adequate volume of Dulbecco's modified Eagle's medium (DMEM)/ high glucose (Sigma-Aldrich, St. Louis, MO, USA) with 10% inactivated serum, aliquoted and stored at −80°C.

The biological titration of lentiviral vectors expressing EGFP was carried out by flow cytometry as previously described [37]. Lentiviral stocks with titers of $1\times10^9$ transducing units per milliliter (TU/ml) were used for experimentation.

## Chicken embryo injection and hatching

Fertilized chicken eggs (Iranian native Golpayegani breed) were obtained from a local poultry farm at the peak production period. Each embryo was injected thrice (Group BAC) at different stages of its development and through different routes as follows: at stage X, into the sub-germinal cavity of blastoderm embryo, at HH stages 14–15, into the dorsal aorta of the embryos, and at HH stage 37 (on the 11th day of incubation), into the CAM vasculature. For the first injection, fresh and fertile eggs were placed vertically with their blunt end down at room temperature (RT) for 4 hours before manipulation, allowing the embryos to float to the highest point inside the shell. A round window with a diameter of approximately 1.5 cm was opened at the pointed end of each egg using a handheld rotary tool (Dremel, model number: 4000-4/65) to reach the blastoderm embryo. Before removing the cut eggshell piece, it was marked according to its position with respect to the cutting edge so that it could easily be put back to its original location. The eggshell pieces were labeled and stored in phosphate buffered saline solution (PBS), containing 10% egg albumen and 1% penicillin-streptomycin and a cell culture incubator at 37°C until further use. Two microliters of the concentrated lentiviral vector stock (S1 Fig) with a titer of $1\times10^9$ TU/ml were injected into the sub-germinal cavity using a pulled-glass capillary needle (inner diameter: 40 μm; World Precision Instruments, Sarasota, FL, USA) attached to an oil hydraulic injection system (CellTram oil, Eppendorf). Before the injection, phenol red (Sigma-Aldrich, St. Louis, MO, USA) was added to the viral solution at a final concentration of 0.1% to facilitate visualization during microinjection. Correct injections were confirmed by visual inspection verifying the spreading of the viral solution across the entire area pellucida but not outside the borders of the area opaca (Fig 1a). After the injection, eggshells were completely filled with fresh albumen collected from other freshly laid eggs (Fig 1b). Having an overflow of egg albumen is necessary to prevent air bubble formation after sealing the window. A small piece of plastic wrap (PVC cling film, Pilgon, Iran) was stretched over the window (Fig 1c) and its ends were secured with a hot glue gun (Shanghai Techway Industrial Co., China) to the eggshell. The sealed eggs were incubated in a vertical position (with the window uppermost) in a humidified incubator (Rcom Maru 380 Deluxe, Rcom, Korea) at 37.7°C and 55% relative humidity without turning.

After 55 h of incubation, the manipulated eggs were removed from the incubator and the plastic wrap was peeled away carefully to avoid damaging the embryo vasculature. Two microliters of the lentiviral vector solution (at the same concentration of the first injection) were

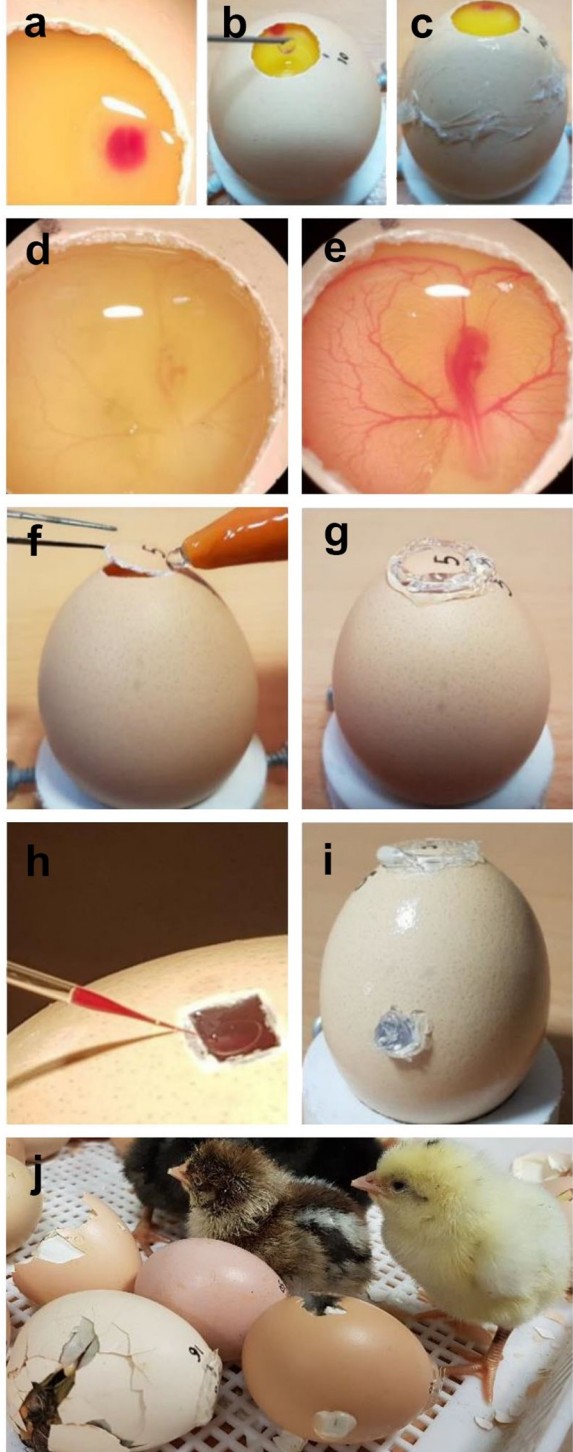

**Fig 1. The three-stage injection method in chicken embryos from the blastoderm stage to the hatching using original eggshells.** (a) A round window with a diameter of approximately 1.5 cm was opened at the pointed end of the non-incubated fertilized eggs (stage X); the viral solution, containing phenol red, was injected into the sub-germinal cavity. The injected solution had to be spread across the area pellucida but it should not surpass the borders of the area opaca. (b) Thin albumen was added to fill the shells so that they would slightly overflow when the cover was put on the window. (c) Plastic wrap was stretched over the window and its ends were secured with a hot glue gun to the eggshell. (d) After 55 h of incubation, the plastic wrap was removed and the second injection was performed into the dorsal aorta at HH stages 14–15 embryo. (e) The distribution of the injection solution, containing phenol red, throughout the

embryonic blood vasculature was done. (f,g) Eggshell fragment was taken out of storage, put back into its respective place across the window, and fixed with a hot glue gun. (h) At 11 days of incubation, a small window was drilled over the site of a blood vessel of the chorioallantoic membrane and before the third injection, the plunger was aspirated to ensure that the needle was inserted into a blood vessel. (i,j) After sealing of the eggshells window with a hot glue gun, the eggs were incubated until hatching.

injected into the dorsal aorta of each embryo at HH stages 14–15 using a pulled-glass capillary needle (inner diameter: 40 μm; World Precision Instruments, Sarasota, FL, USA) attached to an oil hydraulic injection system under a stereo-microscope. In order to visualize and confirm the distribution of the injection solution throughout the embryonic vasculature, phenol red was added to the viral solution at a concentration similar to that of the first injection (Fig 1d and 1e). After the injection, each eggshell fragment was removed from the storage, put back into its respective place across the window, and fixed with a hot glue gun (Fig 1f and 1g). The manipulated eggs were then incubated in an incubator (Rcom Maru 380 Deluxe, Rcom, Korea) at 37.7°C and 55% relative humidity in a vertical position with the air sac uppermost and egg turning every 2 h.

On the $11^{th}$ day of incubation (HH stage 37), the manipulated eggs were candled at their blunt end to identify blood vessels of CAM. A small window was carefully drilled over the site of a prominent blood vessel and the drilled shell piece was gently removed, leaving the underlying white shell membrane intact. A drop of mineral oil was used to render the underlying blood vessel transparent and 4 μl of the viral solution, at the same concentration used previously, were injected through the transparent eggshell membrane into the blood vessels using a pulled-glass capillary needle (inner diameter: 80 μm; World Precision Instruments, Sarasota, FL, USA) attached to an oil hydraulic injection system. Before injection, the plunger was aspirated slightly to ensure that the needle was successfully inserted into a blood vessel (Fig 1h). After the eggshell window was sealed with a hot glue gun (Fig 1i), the eggs were returned to the incubator (Rcom Maru 380 Deluxe, Rcom, Korea). At the end of the $18^{th}$ day of incubation, egg turning was stopped and the eggs were placed in hatching trays at 37.2°C and 65% relative humidity until hatching (Fig 1j).

## Experimental groups

The eggs were grouped according to the route of injection (B: Blastoderm injection at EGK stage X; A: Aorta injection at HH stages 14–15; and C: Chorioallantoic membrane injection at HH stage 37).

Group BAC: a triple injection into the blastoderm, aorta, and CAM
Group BA: a double injection into the blastoderm and aorta
Group BC: a double injection into the blastoderm and CAM
Group AC: a double injection into the aorta and CAM
Group B: a single injection into the blastoderm
Group A: a single injection into the aorta
Group C: a single injection into the CAM
Group Co (Control group): no injection
Group Sh (Sham group): a triple injection of PBS

## Genomic DNA analysis

Genomic DNA samples were extracted from the semen of mature $G_0$ roosters and the blood of $G_1$ hatchlings using the DNeasy Blood and Tissue Kit (Qiagen, Hilden, Germany) according to manufacturer's instructions. For genomic PCR analysis, a primer pair was designed to

include the upper strand (5′- `TCCGATCACGAGACTAGCC` -3′) and lower strand (5′- `CATGGA CGAGCTGTACAA` -3′) sequences and used for amplification of a 780 bp DNA fragment encoding *EGFP* gene. PCR amplification was performed using the subsequent cycles: initial denaturation at 95°C for 3 min, followed by 35 cycles of denaturation at 95°C for 45 sec, primer annealing at 56°C for 45 sec, and polymerization at 72°C for 1 min, followed by a final extension at 72°C for 10 min. The plasmid DNA was used as the positive control, and the genomic DNA isolated from the wild-type chickens was used as the negative control. Finally, 5 μl of each genomic PCR amplicon was analyzed by agarose gel electrophoresis.

### Reverse transcription quantitative PCR (RT-qPCR) analysis

Total RNA was isolated from the testes of $G_0$ newly hatched chicks using TRIzol reagent (Ambion, Austin, TX, USA) according to the manufacturer's instructions. The isolated RNA samples were treated with DNase I (Thermo Scientific, Waltham, MA, USA) and reverse-transcribed using random hexamer-primers and M-MuLV reverse transcriptase (Takara, Japan). The resulted cDNA was used to perform quantitative PCR in an Applied Biosystems Step One Plus real-time PCR system (Applied Biosystems, CA, USA) using SYBR Green PCR Master Mix (Takara, Japan). The relative expression levels of the target gene (*EGFP*) were calculated using the $2^{-\Delta\Delta Ct}$ method. For *EGFP*, the following primers were used for the real-time PCR: forward, 5′- `AGCAGAAGAACGGCATCAAG` -3′ and reverse, 5′- `GGTGCTCAGGTAGTGGTT GTC` -3′. The following primers were also used to amplify chicken glyceraldehydes-3-phosphate dehydrogenase (*GAPDH)* gene as the internal control: forward 5′-`GATTCTACACACGG ACACTTCAAGG` -3′ and reverse 5′-`ACAATGCCAAAGTTGTCATGGATGAC` -3′. All primers designed by Beacon Designer 7.9 software (Premier Biosoft International, Palo Alto, CA, USA). Real-time PCR reactions were performed in triplicates with the following cycling conditions: 95°C for 2 min followed by 40 cycles of 95°C for 30 sec, 55°C for 30 sec, and 72°C for 30 sec.

### Visualization of EGFP in organs of transgenic chickens

The tissue localization of EGFP in transgenic chickens was examined either by the microscopic detection of native EGFP-fluorescence in frozen sections or immunohistochemistry with an anti-GFP antibody in paraffin-embedded sections. Briefly, organ tissues were excised and fixed in 4% paraformaldehyde in PBS at 4°C overnight. For frozen sections, after the overnight cryo-protection of tissues in a sucrose solution (30% sucrose in PBS), the tissues were embedded in OCT compound (Tissue-Tek, Sakura, Torrance, CA). Sections at 7 μm thickness were cut on a cryostat (Leica Biosystems, Wetzlar, Germany), mounted on positively charged slides, and stored at −80°C until assay. For immunohistochemistry staining, the fixed samples were embedded in paraffin and sectioned at 4 μm thickness. After deparaffinization, antigen retrieval and blocking of nonspecific antigenic sites, the sections were incubated with a rabbit anti-GFP antibody (Abcam, Cambridge, UK; 1:1000) overnight at 4°C and then with the goat anti-rabbit Alexa Fluor 488 secondary antibody (Life Technologies, Ltd., Paisley, UK; 1:500) for 2 h at RT, followed by counterstaining with 4′, 6-diamidino-2-phenylindole (DAPI). The immunostained sections were imaged under a fluorescence microscope (BX51; Olympus, Japan), which was equipped with appropriate filter sets and coupled to a digital camera (DP72, Olympus). A negative tissue specimen was used for each tissue type to ensure the elimination of autofluorescence.

### Flow cytometry analysis

Single-cell suspension of testicular cells was prepared by a one-step enzymatic digestion protocol [38]. The resulting single cells were then diluted to $10^6$ cells/ml in PBS and directly

analyzed for EGFP fluorescence on a FACSCalibur instrument using CellQuest software (BD Biosciences, San Jose, CA, USA). EGFP-positive cells were identified by comparison with single-cell suspensions obtained from the testes of wild-type chicks.

## Western blot experiments

Testicular EGFP expression at the protein level was further confirmed by western blot analysis of the whole testis proteins using an anti-GFP antibody. The samples were incubated for 15 minutes at −20˚C after the addition of acetone to allow protein precipitation. Afterwards, the solutions were centrifuged for 10 min at 14,000 g and 4˚C and the precipitates were resuspended in a lysis buffer (7 M urea, 2 M thiourea, 4% CHAPS, 40 mM Tris, pH 8.5, 0.002% bromophenol blue, 65 mM DTT, 1 mM EDTA, and 1 mM PMSF). Equal amounts (30 μg) from each protein sample were separated by SDS-PAGE and transferred to PVDF membranes (PVDF; Biorad, USA). The membranes were blocked with 10% (W/V) skim milk (Merck, USA) and subsequently incubated with primary antibodies for 2 h at RT. Specific primary antibodies were monoclonal purified anti-GFP (Biolegend, San Diego, CA, USA; 1:1000) and monoclonal anti-ß-actin (Clone AC-74, Sigma-Aldrich, St. Louis, MO, USA; 1:500) antibodies. After being washed with TBST buffer (50 mM Tris-HCl, pH 7.6, 150 mM NaCl, and 0.1% tween 20), the membranes were incubated with secondary antibodies for 45 min. The secondary antibodies were horseradish peroxidase (HRP)-conjugated goat anti-mouse IgG (1:5000, Dako, Japan) and HRP goat anti-rat IgG antibody (Biolegend, San Diego, CA, USA; 1:4000). The protein bands were visualized using an Amersham ECL Advance Western Blotting Detection Kit (GE Healthcare, Germany).

## Statistical analysis

Data were analyzed using SPSS software version 23.0 (SPSS Inc., Chicago, IL, USA) by one-way ANOVA, followed by the Tukey's post-hoc test. The hatching rates, hatchling mortality rates, transgenic frequencies, and germline transmission rates of different experimental treatments were compared using the chi-squared test. Values are expressed as mean ± SEM (standard error of the mean) and $P < 0.05$ was considered statistically significant.

## Results

### Effect of injection method on viability and transgenic frequency in $G_0$ presumptive founders

As shown in Table 1, the triple injection (BAC and sham groups) had the most adverse effect on hatchability compared with other types of injection, including single and double injections. Among the injection routes, the lowest hatchability was observed in the blastoderm group (B vs. A and C). However, neither the injection frequency nor the injection route affected the sex ratio. Similar to hatchability, the number of injections and the route of injection also affected the hatchling mortality rate, with the highest rates observed in groups with triple injection and those with injection into the blastoderm. Fig 2a shows DNA amplification (EGFP region) from semen samples collected from $G_0$ roosters. Regarding the transgenic frequency, based on the percentage of EGFP-positive semen samples (Table 1), the lowest frequency was observed when the injection was performed into the blastoderm ($P < 0.05$), whereas no significant difference was observed between the dorsal aorta and CAM route groups ($P = 0.67$). Both the double injection into the dorsal aorta and CAM route (AC group) and the triple injection (BAC group) achieved higher germline-transgenic frequencies in the $G_0$ generation, compared

**Table 1. Hatchability, sex ratio, hatchling mortality, transgenic frequency, and germline transmission efficiency in chimeric chickens produced by different injection routes of viral vector administration.**

| Injection method | Hatchability no. (%) | Male sex ratio in $G_0$ chicks no. (%) | Hatchling mortality no. (%) | EGFP-positive semen samples of G0 roosters no. (%) | Transgenic ratio in $G_1$ progeny no. (%) |
|---|---|---|---|---|---|
| B | 105/160 (65.6%) [a] | 48/105 (45.7%) | 22/105 (20.9%) [c] | 3/24 (12.5%) [a] | 1/220 (0.4%) [a] |
| A | 126/155 (81.2%) [bc] | 54/126 (42.8%) | 11/126 (8.7%) [b] | 11/29 (37.9%) [b] | 17/220 (7.7%) [c] |
| C | 122/140 (87.1%) [be] | 52/122 (42.6%) | 7/122 (5.7%) [ab] | 13/30 (43.3%) [bc] | 6/220 (2.7%) [b] |
| BA | 94/158 (59.5%) [ad] | 49/94 (52.1%) | 18/94 (19.1%) [c] | 15/23 (65.2%) [cd] | 7/220 (3.1%) [b] |
| BC | 102/155 (65.8%) [a] | 49/102 (48.0%) | 19/102 (18.6%) [c] | 18/27 (66.6%) [cd] | 1/220 (0.4%) [a] |
| AC | 121/160 (75.6%) [c] | 59/121 (48.7%) | 12/121 (9.9%) [b] | 20/29 (68.9%) [d] | 30/220 (13.6%) [d] |
| BAC | 97/178 (54.5%) [d] | 51/97 (52.5%) | 21/97 (21.6%) [c] | 19/25 (76.0%) [d] | 21/220 (9.5%) [cd] |
| Sh | 82/155 (52.9%) [d] | 36/82 (43.9%) | 16/82 (19.5%) [c] | 0 | 0 |
| Co | 127/140 (90.7%) [e] | 59/127 (46.4%) | 3/127 (2.3%) [a] | 0 | 0 |

A, B, C: Viral vector injection into blastoderm (B), dorsal aorta (A), and chorioallantoic membrane vasculature (C). Co: Control group with no manipulation; Sh: Sham group with PBS injection into the B, A, and C sites. Hatchability (number of hatched chicks/injected eggs). Values with different letters in the same column are significantly different ($P < 0.05$).

to the single injection groups ($P < 0.05$), while no significant difference was found between the double- and triple-injection groups ($P > 0.41$).

## RT-qPCR and flow cytometry results

We quantitatively assessed the EGFP expression at both mRNA (Fig 2b) and protein (Fig 3) levels in the testes of $G_0$ newly hatched chicks. The RT-qPCR data revealed that the triple-injection group had the highest EGFP expression, which was significantly higher than that of

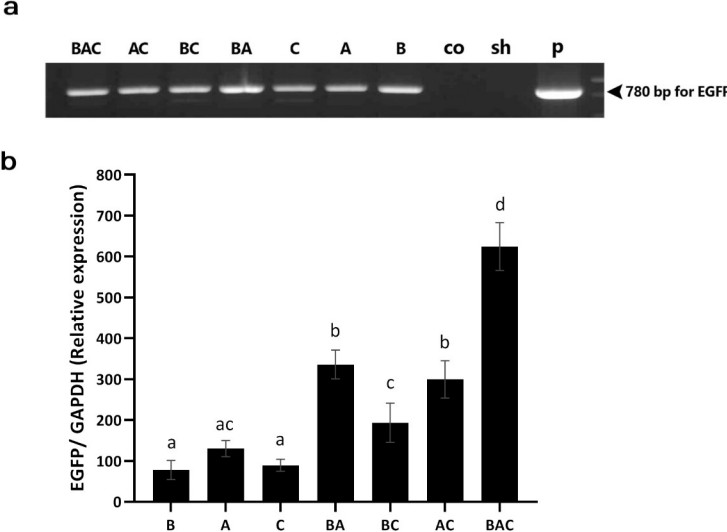

**Fig 2. PCR screening of putative transgenic male founders.** (a) Genomic PCR analysis of the semen of $G_0$ roosters. The EGFP reporter gene in the semen samples of $G_0$ roosters in all vector-injected groups was detected by PCR. DNA samples used as templates for PCR were as follows: Plasmid DNA (pWPXL) (P) as positive control, genomic DNA extracted from the semen of the seven vector-injected (left to right lanes, respectively), control (Co) and sham (Sh) groups. (b) Quantitative RT-PCR analysis of testis mRNA from $G_0$ chickens. Total mRNA was isolated from the testicular tissues of $G_0$ chickens and analyzed by RT-PCR. The data were normalized with the expression of chicken GAPDH.

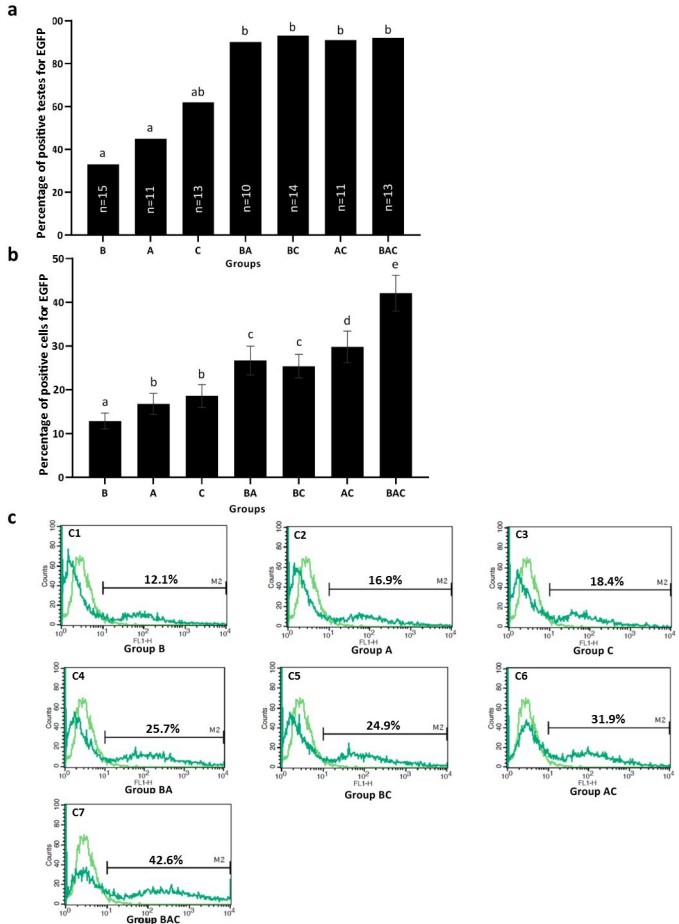

**Fig 3. The testicular expression of EGFP at protein level was confirmed by flow cytometry in $G_0$ chickens.** (a) The frequencies of EGFP positivity based on flow cytometry analysis in samples taken from the vector-injected groups. $n$ = number of testes evaluated by flow cytometry in each group. (b) and (c) Testicular cells were harvested by trypsinization, and approximately $5 \times 10^5$ cells/500 μl PBS were investigated by a BD FACS Calibur flow cytometer, using the testicular cells of non-transgenic chicks as negative controls. The relative cell number counts are plotted as a function of variable intensities of the green fluorescence from individual cells. Pale green line graph: Non-transgenic cells; dark green line graph: Transgenic cells. The percentage of total EGFP-positive cells is shown in each histogram. (b) The positive cell percentages were plotted as mean ± SEM of the data from five independent experiments. Values with different letters are significantly different ($P < 0.05$).

the double- and single-injection groups ($P < 0.05$) (Fig 2b). Among the double-injection groups, the relative expression of EGFP was significantly lower in the BC group than BA and AC groups ($P < 0.05$) (Fig 2b). Among double- and single-injection groups, only two groups (BC and A) showed no significant difference ($P > 0.05$) (Fig 2b). No significant differences were observed among the single-injection groups ($P > 0.05$) (Fig 2b).

As depicted in Fig 3a, over 90% of the testis samples which were collected from the double- and triple-injection groups and analyzed by flow cytometry contained EGFP positive cells, while this percentage drastically reduced in testis samples collected from the single injection groups, 33.3% (B), 45.4% (A), and 61.5% (C). The results of flow cytometry analysis also revealed that the percentage of testicular cells positive for EGFP was significantly higher in the triple-injection group (BAC) than that of the double injection groups ($P < 0.05$) and in the double injection groups than that of the single injection groups ($P < 0.05$) (Fig 3b and 3c). Among the double injection groups, no significant difference was observed between BA and

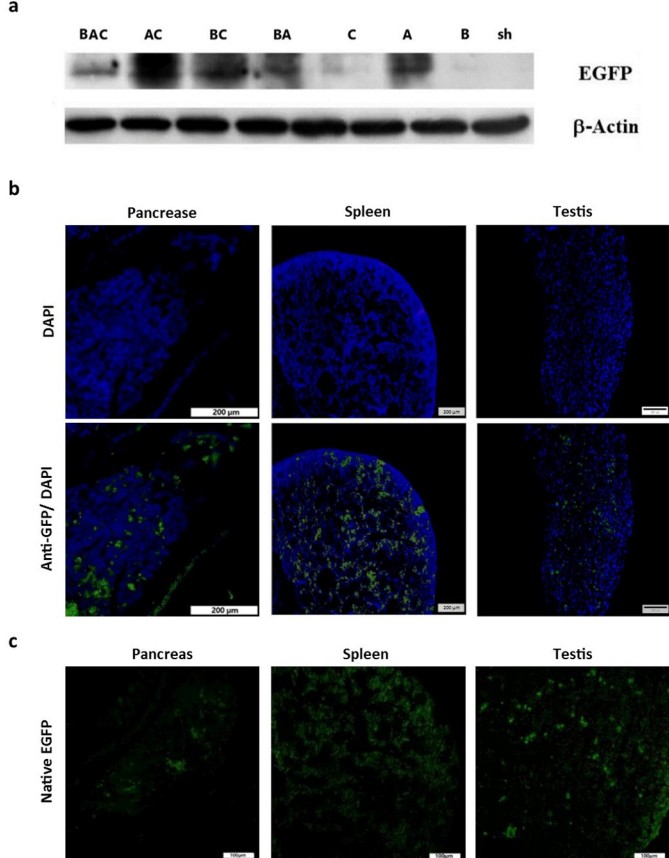

**Fig 4. Detection of EGFP protein using Western blotting and fluorescence microscopy in $G_0$ chickens.** (a) Western blotting analysis of the testicular protein extracts from the seven vector-injected groups and the sham group. $\beta$-actin was used as the loading control. (b) The mosaic expression of EGFP was confirmed in the pancreas, spleen and testis tissue sections from a $G_0$ chick in group C by use of Alexa-Fluor488 conjugated anti-GFP antibody stained on paraffin-embedded tissues. (c) Direct detection of EGFP fluorescence in OCT-embedded tissues from a $G_0$ chick in group C.

BC groups ($P > 0.05$); however, the percentages of EGFP positive testicular cells in the afore-mentioned groups were significantly lower than that of the AC group ($P < 0.05$) (Fig 3b and 3c). We also detected no significant difference between groups A and C in terms of the per-centage of EGFP positive cells ($P > 0.05$), but the mean values for these two groups were signif-icantly higher than that of the group B ($P < 0.05$) (Fig 3b and 3c).

## Western blot analysis and fluorescence microscopy

We assessed the EGFP and $\beta$-actin expression at the protein level by Western blot analysis in the testicular tissues of $G_0$ hatched chicks (Fig 4a) in which their EGFP expression was previ-ously confirmed by the RT-qPCR (Fig 2b) and flow cytometry (Fig 3) assays. Our results showed that all the seven vector-injected groups were positive for EGFP protein. Despite simi-lar levels of $\beta$-actin expression, the level of EGFP expression varied between groups. Fig 4b shows the immunostaining for EGFP protein in the pancreas, spleen and testis tissue sections from a $G_0$ chick in group C, indicating mosaic expression of the transgene in these tissues. The assessment of the EGFP expression in cryosections of different tissues showed that native EGFP fluorescence was present in most of the examined tissues; as shown in the pancreas, spleen and testis tissues taken from a $G_0$ chick in group C (Fig 4c).

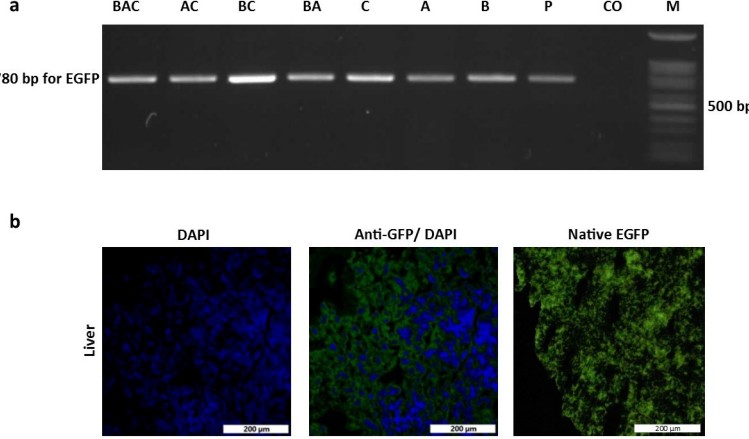

**Fig 5. Identification of G₁ transgenic chicks expressing EGFP.** (a) PCR analysis was performed on blood DNA from G₁ chicks. DNA samples used as templates for PCR were as follows: Plasmid DNA (pWPXL) (P) as positive control, genomic DNA extracted from the blood of the G₁ progenies of the seven vector-injected (left to right lanes, respectively) and control (Co) groups. Lane M, DNA marker. (b) Ubiquitous EGFP fluorescence is shown in the liver tissue of a transgenic G₁ chick.

## Germline transmission evaluation in G₁ generation

To assess the germline transmission (GT) efficiency, three germline chimeric roosters from each group were randomly selected and mated with wild-type hens. DNA derived from the blood of G₁ chicks, obtained by the above crossing, was assessed for the presence of EGFP using PCR (Fig 5a), and the results were expressed as transgenic ratios in G₁ progeny (Table 1). The best result for this parameter was observed in the AC group, which received a double injection into the dorsal aorta at HH stages 14–15 and CAM, with a GT efficiency of 13.6%. Although the ratio for the triple-injection group (BAC) was lower than that of the AC group, the difference was not significant ($P = 0.18$). Comparison of the results between the three independent injection routes revealed that the lowest rate of GT was achieved when the injection was carried out into the blastoderm (0.4%), whereas injection into the dorsal aorta at HH stages 14–15 resulted in the highest GT rate (7.7%). Injection of viral vectors into the CAM vasculature resulted in a GT efficiency of 2.7%, which was significantly higher than that obtained by injection into embryos at the blastoderm stage ($P < 0.05$). Transgene inheritance in G₁ chicks was further confirmed by the sequencing of PCR products obtained from the amplification of blood DNA (S2 Fig). Ubiquitous EGFP fluorescence was detected in various tissues of transgenic G₁ chicks, as shown for the liver tissue in Fig 5b.

## Discussion

A variety of strategies have been used for the generation of transgenic poultry; however, the efficiency of chimera production and transgene inheritance is still low [5]. To improve the efficiency of generating germline-transmitting chimeric chickens, we aimed to assess the multi-injection of viral vectors into embryos at three different stages of development using the advantages of HIV-1 derived lentiviral vectors (S1 Fig) and modified manipulation of chicken eggs at the desired developmental stages (Fig 1). In this modified approach, we used the original eggshells instead of the surrogate eggshells [39] to maintain embryo development from laying eggs to hatching. Up to our knowledge, this is the first report presenting this novel modified approach for improving germline transmission efficiency in chicken chimeras.

The minimum hatchability of this technique was 52.9%, which is higher than that reported (less than 37%) in most of the previous studies using a surrogate eggshell system [6,11,13,15,18,22,40–43]. Although some improvements have been reported in the culture of chicken embryos [44–47], culture methods using surrogate eggshells or artificial vessels suffer from several disadvantages, such as high economic costs, long operating time, inconsistency between the morphology of the original eggs and the surrogate shells, and the risk of injury to or contamination in the embryo during the process of transferring egg content [48]. In addition, it is important to note that the rates of hatchability achieved in this study is following single-, double- and triple- injection and it is clear that over-manipulation could be more detrimental to embryo survival than a single manipulation. Therefore, the new approach described in this study could have potentially important consequences for future research on the production of recombinant proteins in highly efficient poultry systems rather than other expression systems such as bacteria or mammalian cells [21,49].

The prevention of the transfer of embryos from their shells to surrogate shells can be considered as one of the reasons for the improved efficiency of the modified approach in this study. Other reasons for this improved efficiency could be the reduced dead-in-shell mortality rate, which routinely occurs during the last 3 days of incubation in the classical surrogate eggshell system due to yolk sac infection and egg turning limitation [46,48], and also maintaining optimal gas exchange across the eggshell pores during incubation by keeping the developing embryos in their original eggshells. It is imperative to underscore that the aim of this study was not to compare two methods; rather, the study was designed to investigate the efficiency of generating germline transgenic chickens using multiple injections at three different developmental stages.

The tissue composition and accessibility of the CAM for experimental manipulation, have attracted many researchers. The highly vascularized network of the CAM has been widely used as an experimental model to evaluate tumor cell metastasis and dissemination *in vivo*, as well as to study anticancer drug effects on tumor inhibition [50–53]. Several studies have demonstrated that many tumor cells can be metastasized into various tissues (e.g. liver, spleen, lung, pancreas, brain, bursa of fabricius, bone marrow, ovary, etc.) after cell inoculation or intravasation into the CAM vasculature [53–57]. Hen and colleagues [58] were the first researchers who used the CAM as a route of gene delivery into the somatic tissues of chickens with the highest levels of expression in the liver and spleen. In the current study, for the first time, we demonstrated the possibility of using CAM as an accessible route of gene transfer to germline cells. However, the mechanistic basis of the above observations and our findings remains unclear due to the lack of enough knowledge about the developmental characteristics of CAM such as its lymphatic and vascular system remodeling and permeability [52], which could affect the pattern of biodistribution of viral vector particles. Based on the pilot experience, we found that a minimum volume of 4 μl of concentrated viral stock ($1 \times 10^9$ TU/ml) had to be injected into the CAM for consistent detection of *EGFP* gene in the semen samples. However, due to the limited capacity of the receiving vessels, the amount injected into the dorsal aorta of embryos at HH stages 14–15 did not exceed 2 μl. Therefore, caution should be exercised when comparing the two injection routes in terms of GT efficiency as similar volumes of viral vectors were not injected. We acknowledge that this may be seen as a shortcoming of our approach, but improving the overall GT efficiency was the main focus of this study.

Germline chimeric roosters ($G_0$) which were identified as positive based on the genomic PCR screen were mated with wild type hens in order to assess the rates of GT of the transgene to the resultant $G_1$ offspring. We didn't measure the level of chimerism for each individual $G_0$ rooster, because the main focus of this study was to assess the overall germline transgenesis for each method. However, we acknowledge this as a shortcoming of our analysis, making it

impossible to assess the correlation between the degree of chimerism in each selected $G_0$ rooster and the level of germline transgenesis. PCR analysis was performed on blood DNA from $G_1$ chicks to identify transgenic individuals. Subsequently, DNA sequencing of the PCR products was carried out to confirm transgene integration in the $G_1$ chicks. The injection of lentiviral vectors into the dorsal aorta at HH stages 14–15 yielded a GT efficiency of 7.7%, which was significantly higher than those when the injection was administered into the blastoderm (0.4%) or CAM (2.7%). Despite the significant improvement observed in the GT efficiency when using the dorsal aorta and CAM routes compared to the blastoderm route, in the double-injection system, when the injection was administered into the blastoderm and then either dorsal aorta (BA: GT efficiency 3.1%) or CAM (BC: GT efficiency 0.4%), the GT efficiency was significantly reduced. These results indicate that the injection into the blastoderm is inversely correlated with the GT efficiency. Similar results were observed when the triple-injection group (BAC: GT efficiency 9.5%) was compared with the double-injection group (AC: GT efficiency 13.6%), further emphasizing the inverse correlation mentioned above. This observation raises the possibility that early injection into chick embryos at blastoderm stage may lead to a decrease in germ cells pool which is considered as the target of gene transfer at later stages.

The results regarding the EGFP expression at the mRNA level were not totally in line with the above findings. The relative expression of EGFP at the mRNA level was similar between the groups with a single injection despite to the different injection routes. However, in the double- and triple- injection groups, the relative expression of EGFP at the mRNA level was significantly higher than that of the single injection groups (Fig 2b). These findings indicate that although the mRNA expression level was higher in the BA, BC, and BAC groups, which received an additional injection into the blastoderm, their GT efficiency did not increase compared to A, C, and AC groups, respectively. This incompatibility may be related to the differential ability of PGC in the blastoderm, compared to that of the circulating PGC at HH stages 14–15 or gonadal germ cells at the HH stage 37, to allow for transgene integration into their genome. This differential ability could be attributed to the high heterochromatin content and chromatin epigenetic modifications in early PGC in the blastoderm relative to the late PGC [59]. It is also of note that the number of PGC at the blastoderm stage was reported to be significantly lower compared to the later stages [60]. In addition, PGC at the blastoderm stage might be more vulnerable to manipulation than those at the later stages. These two latter possibilities can also account for the differential gene targeting ability of PGC in the blastoderm. Furthermore, the increase in the EGFP mRNA expression could be related to other testicular cells as the efficiency of GT is affected by the efficiency of gene targeting in PGC. Another possibility is that although the PGC in the blastoderm were targeted successfully by the viral vectors, they were not able to form functional gametes that can produce offspring. Similar results were also observed at the protein level assessed by flow cytometry (Fig 3b and 3c). The percentage of testis samples positive for EGFP also indicates that the injection into the blastoderm is not the best route to obtain transgene-expressing testis cells with high repeatability (Fig 3a). The finding of low germline transgenesis observed in this study was consistent with that of the previous studies which used blastoderm-stage embryos for germline gene transfer [14,15,17–21,25]. The results of this study and those of earlier studies suggest that early blastoderm PGC in chick embryos may not be a suitable target for transgene integration and hence gene targeting in chicken germ cells should be aimed at later stages [23,25,61–64].

The groups receiving at least one injection into the blastoderm, including the B, BA, BC, BAC, and Sh groups, showed the highest $G_0$ hatchling mortality and the lowest hatchability rates. These results might be due to the direct effect of early viral injection on embryo survival

and post-hatching development or, alternatively, the effect of air entrance on the embryo survival during the blastoderm stage associated with egg windowing [65].

## Conclusions

The results of this study showed that the embryo's bloodstream at HH stages 14–15 was the most effective route of viral vector administration in terms of germline gene transfer. The embryo CAM vasculature was successfully used as a novel route of gene transfer into germline tissues. Thus, CAM can be considered as a possible route for the germline transmission of *in vitro* modified PGC in future studies. In addition, double injection of transgenes into the dorsal aorta, at HH stages 14–15, and CAM vasculature, at HH stage 37, was found to be the most efficient method in terms of hatchability rates of the manipulated eggs, hatchling viability and GT efficiency. This new double-injection strategy combines the simplicity and reliability, so that can be employed in place of the classical less efficient method of using a single injection into the sub-germinal cavity of blastoderm embryos. Accordingly, we believe that this new strategy has the potential to become a versatile and effective tool for germline gene transfer.

## Supporting information

**S1 Fig. The schematic representation of the pWPXL lentiviral vector.** LTR, long terminal repeat; RRE, ref-responsive element; cPPT, central polypurine tract; pEF1a, human elongation factor-1 alpha promoter; EGFP, enhanced green fluorescent protein; WPRE, woodchuck hepatitis virus posttranscriptional control element.
(PDF)

**S2 Fig. A sample sequencing of the genomic PCR product that contained the EGFP sequence amplified from a genomic sample of transgenic $G_1$ chick.** Here, the sequence of genomic PCR product is compared with the EGFP sequence in pWPXL viral vector using SnapGene software. The result of sequence analysis confirmed the integration of the vector into the chicken genome. Upper sequence: EGFP ORF amplified from the genome-integrated pWPXL. Lower sequence: EGFP ORF in pWPXL viral vector.
(PDF)

**S1 Raw images.**
(PDF)

## Author Contributions

**Conceptualization:** Danial Naseri, Mohammad Hossein Nasr-Esfahani.

**Data curation:** Danial Naseri, Kianoush Dormiani.

**Formal analysis:** Danial Naseri.

**Funding acquisition:** Mohammad Hossein Nasr-Esfahani.

**Investigation:** Danial Naseri.

**Methodology:** Danial Naseri, Kianoush Dormiani, Mehdi Hajian, Farnoosh Jafarpour, Mahboobeh Forouzanfar, Naeimeh Karimi.

**Project administration:** Danial Naseri.

**Supervision:** Danial Naseri, Kianoush Dormiani, Mohammad Hossein Nasr-Esfahani.

**Validation:** Danial Naseri, Kianoush Dormiani, Mohammad Hossein Nasr-Esfahani.

**Visualization:** Danial Naseri.

**Writing – original draft:** Danial Naseri, Farnoosh Jafarpour, Mohammad Hossein Nasr-Esfahani.

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
