## [Decision Letter · Decision Letter 0]

22 Mar 2021

PONE-D-21-04208

Improving germline transmission efficiency in chimeric chickens using a multi-stage injection approach

PLOS ONE

Dear Dr. Nasr-Esfahani,

Thank you for submitting your manuscript to PLOS ONE. After careful consideration, we feel that it has merit but does not fully meet PLOS ONE’s publication criteria as it currently stands. Therefore, we invite you to submit a revised version of the manuscript that addresses the points raised during the review process.

We look forward to receiving your revised manuscript.

Kind regards,

Xiuchun Tian

Academic Editor

PLOS ONE

Journal Requirements:

Additional Editor Comments (if provided):

Reviewers' comments:

Reviewer's Responses to Questions

**Comments to the Author**

1. Is the manuscript technically sound, and do the data support the conclusions?

Reviewer #1: Yes

Reviewer #2: Yes

2. Has the statistical analysis been performed appropriately and rigorously? 

Reviewer #1: N/A

Reviewer #2: Yes

3. Have the authors made all data underlying the findings in their manuscript fully available?

Reviewer #1: Yes

Reviewer #2: Yes

4. Is the manuscript presented in an intelligible fashion and written in standard English?

Reviewer #1: Yes

Reviewer #2: Yes

5. Review Comments to the Author

Reviewer #1: Genetic modification of Avian species is not straightforward due to the special location of the ovum. During the past decades a variety of solutions it has been published. Scientific advances in recent years have resulted that it is no longer a problem in any laboratory to establish and maintain avian PG cell lines. The exact protocol of this is described in the handbook “Reproductive Technologies in Animals, Chapter Reproductive Technologies in Avian Species (2020). Therefore, lines in (Introduction Line 76-79) should be deleted. Currently, the most commonly used method for genetic modification in birds is the gene editing of PG cells with the CRISPR - Cas9 system. However, it is not suitable for the introduction of large sized gene sequences. Therefore, I accept that the authors used a lentivirus as a vector.

The studies and the experimental work presented in the manuscript are well designed and thorough, as well as contain a lot of invested work.

The main problem, apart from the small details, is with the part of the methodological innovation that the authors use two windows to inject into the dorsal aorta and the CAM system. It would be sufficient to open a window on the lateral side of the egg and use it twice for injection. As each new window is opened, the hatching rate is decreases (Discussion Line 398-399). Currently most of the laboratories open the window for injection into the aorta in the lateral side of the egg, although indeed very few publications describe it exactly (Reproductive Technologies in Animals, Chapter Reproductive Technologies in Avian Species (2020) Ed.: Presicce, page 218.). Zhang et al, together with other studies, compare the efficiency of the two window opening methods and find that the efficiency of the lateral side is better! Sztán et al (2017) are used it on goose species.

Therefore, I think that the “Conclusion” and “Discussion” sections should be reworded by the Authors so also address the possibility that the two injections (AC: the best combination) can be accomplished by opening only one window. I think that would be required to do such investigation additionally.

As the opening of the window on the egg is not an innovation, it is at most the opening of the two windows, which is completely unnecessary; therefore the Abstract Lines 27 to 28 („We established a new method of eggshell windowing for embryo manipulation at different developmental stages.”) should be deleted from the Abstract.

Minor deficiencies: In the Material and Method section, some details remained unclear.

Line 17: „handheld rotary tool”; type, manufacturer or photo of the device

Line 109: Exactly eggs of what type of chicken breeds did they work? What does it mean “wild type” (Results Line 356.)?

Line 130: “plastic wrap”; type, manufacturer

Line 131, 160, and 171: “hot glue”? What exactly was the glue used to cover the window? With a hot glue gun? Type, manufacturer?

Line 155, 168, and 170: „glass needle” Did they mean microcapillary? Or Pasteur pipette? How was the liquid moved in it? Also with the “oil hydraulic system”?

Line 161, 171: “incubator” type and manufacturer are missing.

These may seem like unnecessary information, but without knowing them, the experiment cannot be repeated!

References for windowing technique:

1. Reproductive Technologies in Animals, Chapter Reproductive Technologies in Avian Species (2020) Ed.: Giorgio Presicce.

2. Zhang, Y. et al: Isolation of chicken embryonic stem cell and preparation of chicken chimeric model. Molecular Biology Reports 40(3) 2012. DOI: 10.1007/s11033-012-2274-8.

3. Sztán, N. et al: Successful chimera production in the Hungarian goose (Anser anser domestica) by intracardiac injection of blastodermal cells in 3-day-old embryos. Reproduction, Fertility and Development 29(11) 2206-2216. 2017. https://doi.org/10.1071/RD16289.

Other missing references for Introduction or Discussion section:

1. Motono, M., Yamada, Y., Hattori, Y., Nakagawa, R., Nishijima, K., Iijima, S.: Production of transgenic chickens from purified primordial germ cells infected with a lentiviral vector. Journal of Bioscience and Bioengineering, Volume 109, Issue 4, April 2010, Pages 315-321.

2. Sang, H.M.: Genetic modification of the chicken: New technologies with potential applications in poultry production. Biology of Breeding Poultry. Volume 29, 28 May 2009, Pages 45-53.

Reviewer #2: This manuscript describes a comparison of methods for generating germline modified chickens using a lentivirus vector. The manuscript is very well written and easily understood. The authors also provide a very thorough overview of the field.

The results compare three routes of embryo injection: blastoderm (pre-incubation), aorta (day 2.5) and CAM (day 11). The authors tested single methods of injection plus all combinations. They show that aorta injection is the best, however they observed an increase in germline transgenesis when they also injected into the CAM. This is a very surprising result as there are no circulating PGCs in the embryo at this stage, so the lentivirus somehow enters the gonad and infects PGCs (or probably differentiated PGCs). CAM injection alone also gave 2.7% germline transgenesis in G1 progeny. The authors comment in the discussion that this is surprising, and the mechanistic basis of this finding remains unclear. I think the manuscript would benefit from further speculation in the discussion about this surprising result.

Germline transmission evaluation was done by selecting three chimeric G0 roosters at random and placing in matings with wild type hens. Were all three males placed with a pool of hens or were they individually penned? If so, please provide the data for each individual rooster. Also please provide semen analysis on the level of chimerism for each selected rooster. This can be easily measured by PCR on genomic DNA preps from three semen samples collected two weeks apart for each rooster. This data will help readers to understand if the measured level of germline transgenesis for each compared method is consistent across all three selected roosters or possibly only being contributed by a single rooster. I think it would have been a better experimental design to measure chimerism in all G0 roosters and select the three with the highest levels to place in wildtype matings.

6. PLOS authors have the option to publish the peer review history of their article (what does this mean?). If published, this will include your full peer review and any attached files.

Reviewer #1: No

Reviewer #2: No

---

## [Author Response · Author response to Decision Letter 0]

24 Apr 2021

Editor's comments;

We note that you have included the phrase “data not shown” in your manuscript. Unfortunately, this does not meet our data sharing requirements. PLOS does not permit references to inaccessible data. If the data are not a core part of the research being presented in your study, we ask that you remove the phrase that refers to these data. 

Response: We removed the phrase “data not shown” in the revised manuscript according to the journal's requirements.

PLOS ONE now requires that authors provide the original uncropped and unadjusted images underlying all blot or gel results reported in a submission’s figures or Supporting Information files.

Response: We attached the original uncropped and unadjusted images underlying western blot and gel results.

Reviewers' comments;

Reviewer #1: Genetic modification of Avian species is not straightforward due to the special location of the ovum. During the past decades a variety of solutions it has been published. Scientific advances in recent years have resulted that it is no longer a problem in any laboratory to establish and maintain avian PG cell lines. The exact protocol of this is described in the handbook “Reproductive Technologies in Animals, Chapter Reproductive Technologies in Avian Species (2020). Therefore, lines in (Introduction Line 76-79) should be deleted. Currently, the most commonly used method for genetic modification in birds is the gene editing of PG cells with the CRISPR - Cas9 system. However, it is not suitable for the introduction of large sized gene sequences. Therefore, I accept that the authors used a lentivirus as a vector.

Response: Thank you for your comment. We agree that the PGC-mediated genome editing is the most attractive method used for genetic modification in avian species, but the isolation, culture, and genome modification of PGC, as well as screening of genome-edited PGC, are long and complex processes. In addition, this method is currently limited to chickens as optimal conditions for PGC culture isolated from different avian species have to be established. However, we acknowledge that this method is well-established in chickens and is no longer a challenge. So, we reworded the Introduction Line 76-79 as follows: “However, the processes of isolation, culture, and genetic modification of PGC, as well as screening of genome-edited PGC, are time-consuming and complicated procedures for routine use which require highly skilled operators”. We hope that this modification would be acceptable.

Reviewer #1: The studies and the experimental work presented in the manuscript are well designed and thorough, as well as contain a lot of invested work. The main problem, apart from the small details, is with the part of the methodological innovation that the authors use two windows to inject into the dorsal aorta and the CAM system. It would be sufficient to open a window on the lateral side of the egg and use it twice for injection. As each new window is opened, the hatching rate is decreases (Discussion Line 398-399). Currently most of the laboratories open the window for injection into the aorta in the lateral side of the egg, although indeed very few publications describe it exactly (Reproductive Technologies in Animals, Chapter Reproductive Technologies in Avian Species (2020) Ed.: Presicce, page 218.). Zhang et al, together with other studies, compare the efficiency of the two window opening methods and find that the efficiency of the lateral side is better! Sztán et al (2017) are used it on goose species. Therefore, I think that the “Conclusion” and “Discussion” sections should be reworded by the Authors so also address the possibility that the two injections (AC: the best combination) can be accomplished by opening only one window. I think that would be required to do such investigation additionally.

Response: We carried out the same procedure of eggshell windowing for injection at each stage of development because we wanted to keep the egg handling conditions constant in all experimental groups. Since injection into the blastoderm embryo requires the opening of the eggshell at the pointed end of the egg (for technical reasons) and because the timing of this injection is earlier than the timing of the Aorta injection at HH stages 14-15, we used the same window for injection into Aorta. On the other hand, injection into CAM vasculature requires the eggs to be candled just before opening the eggs in order to identify a prominent blood vessel of the CAM. So, by attention to this fact that at early stages of development (HH stages 14-15) CAM vasculature are not yet formed, it's impossible to determine the exact location of the eggshell under which a prominent blood vessel of the CAM will form at later stages. It is of note that injection into CAM vasculature is possible just by making a very small window in the eggshell without removing the underlying shell membrane and this type of egg manipulation didn’t adversely affect the hatchability rates significantly (as shown in table 1). At Discussion Line 398-399 we mentioned that “over-manipulation could be more detrimental to embryo survival than a single manipulation” and by using the word “over-manipulation” we mean the over-manipulation of the egg and embryo and not just necessarily the eggshell windowing. We hope that our reasoning is acceptable by the respected reviewer. 

Reviewer #1: As the opening of the window on the egg is not an innovation, it is at most the opening of the two windows, which is completely unnecessary; therefore, the Abstract Lines 27 to 28 („We established a new method of eggshell windowing for embryo manipulation at different developmental stages.”) should be deleted from the Abstract. 

Response: Thanks, according to the reviewer comment, the sentence was deleted from the Abstract.

Line 117: „handheld rotary tool”; type, manufacturer or photo of the device 

Response: Thanks for the comment. We included the requested information (manufacturer and model number) in the revised manuscript in line 119.

Line 109: Exactly eggs of what type of chicken breeds did they work? What does it mean “wild type” (Results Line 356.)?

Response: The breed of chickens from which fertilizes eggs were obtained was included in the revised manuscript in line 111. The “wild type” is the typical form of a species (not manipulated in a laboratory/non-transgenic) as it occurs in nature.

Line 130: “plastic wrap”; type, manufacturer?

Response: We included the requested information (type, manufacturer) in the revised manuscript in line 134.

Line 131, 160, and 171: “hot glue”? What exactly was the glue used to cover the window? With a hot glue gun? Type, manufacturer?

Response: We used a hot glue gun to seal the eggshell window. It was rectified in the revised version (Line 135, 147, 151, 154, 166, 179) and the device manufacturer was also included (Line 135).

Line 155, 168, and 170: „glass needle” Did they mean microcapillary? Or Pasteur pipette? How was the liquid moved in it? Also with the “oil hydraulic system”?

Response: by “glass needle” we mean “glass capillary needle”. The injection solution was aspirated into the needle by using an oil hydraulic injection system. It was rectified in the revised version (Line 125-127, 160-162, 175-177).

Line 161, 171: “incubator” type and manufacturer are missing.

Response: Thanks for the attention of the reviewer. It was included in the revised version (Line 167, 179).

Reviewer #2: Germline transmission evaluation was done by selecting three chimeric G0 roosters at random and placing in matings with wild type hens. Were all three males placed with a pool of hens or were they individually penned? If so, please provide the data for each individual rooster. Also please provide semen analysis on the level of chimerism for each selected rooster. This can be easily measured by PCR on genomic DNA preps from three semen samples collected two weeks apart for each rooster. This data will help readers to understand if the measured level of germline transgenesis for each compared method is consistent across all three selected roosters or possibly only being contributed by a single rooster. I think it would have been a better experimental design to measure chimerism in all G0 roosters and select the three with the highest levels to place in wildtype matings.

Response: We appreciate the comment of the reviewer. Since in group B only three G0 roosters were identified as germline chimeras (table 1), and hence we had no choice but to choose from either of them, all of these three roosters were allowed to mate with wild type hens in one place. Accordingly, in order to keep the sampling conditions constant, three chimeric roosters from each of the other groups were randomly selected and mated with wild-type hens. We didn't measure the level of chimerism for each individual rooster, because the main focus of this study was to assess the overall germline transgenesis for each method. However, we acknowledge that the comment of the respected reviewer raises a valid point regarding the correlation between the degree of chimerism in each selected G0 rooster and the level of germline transgenesis. We included this point as a shortcoming of our work in the revised version in Line 445-449, and we will try to consider this suggestion in our future studies.

---

## [Decision Letter · Decision Letter 1]

19 May 2021

Improving germline transmission efficiency in chimeric chickens using a multi-stage injection approach

PONE-D-21-04208R1

Dear Dr. Nasr-Esfahani,

We’re pleased to inform you that your manuscript has been judged scientifically suitable for publication and will be formally accepted for publication once it meets all outstanding technical requirements.

Kind regards,

Xiuchun Tian

Academic Editor

PLOS ONE

Additional Editor Comments (optional):

Reviewers' comments:

Reviewer's Responses to Questions

**Comments to the Author**

1. If the authors have adequately addressed your comments raised in a previous round of review and you feel that this manuscript is now acceptable for publication, you may indicate that here to bypass the “Comments to the Author” section, enter your conflict of interest statement in the “Confidential to Editor” section, and submit your "Accept" recommendation.

Reviewer #1: All comments have been addressed

2. Is the manuscript technically sound, and do the data support the conclusions?

Reviewer #1: Yes

3. Has the statistical analysis been performed appropriately and rigorously? 

Reviewer #1: Yes

4. Have the authors made all data underlying the findings in their manuscript fully available?

Reviewer #1: Yes

5. Is the manuscript presented in an intelligible fashion and written in standard English?

Reviewer #1: Yes

6. Review Comments to the Author

Reviewer #1: (No Response)

7. PLOS authors have the option to publish the peer review history of their article (what does this mean?). If published, this will include your full peer review and any attached files.

Reviewer #1: No

---

## [Editor Report · Acceptance letter]

24 May 2021

PONE-D-21-04208R1 

­­Improving germline transmission efficiency in chimeric chickens using a multi-stage injection approach 

Dear Dr. Nasr-Esfahani:

I'm pleased to inform you that your manuscript has been deemed suitable for publication in PLOS ONE. Congratulations! Your manuscript is now with our production department. 

Kind regards, 

on behalf of

Dr. Xiuchun Tian 

Academic Editor

PLOS ONE